# $\mathcal{L}_{\mathrm{DMI}}$: A Novel Information-theoretic Loss Function for Training Deep Nets Robust to Label Noise

**Yilun Xu**[*]**, Peng Cao**[*]

School of Electronics Engineering and Computer Science, Peking University
{xuyilun,caopeng2016}@pku.edu.cn

**Yuqing Kong**
The Center on Frontiers of Computing Studies,
Computer Science Dept., Peking University
yuqing.kong@pku.edu.cn

**Yizhou Wang**
Computer Science Dept., Peking University
Deepwise AI Lab
Yizhou.Wang@pku.edu.cn

## Abstract

Accurately annotating large scale dataset is notoriously expensive both in time and in money. Although acquiring low-quality-annotated dataset can be much cheaper, it often badly damages the performance of trained models when using such dataset without particular treatment. Various methods have been proposed for learning with noisy labels. However, most methods only handle limited kinds of noise patterns, require auxiliary information or steps (*e.g.*, knowing or estimating the noise transition matrix), or lack theoretical justification. In this paper, we propose a novel information-theoretic loss function, $\mathcal{L}_{\mathrm{DMI}}$, for training deep neural networks robust to label noise. The core of $\mathcal{L}_{\mathrm{DMI}}$ is a generalized version of mutual information, termed Determinant based Mutual Information (DMI), which is not only information-monotone but also relatively invariant. *To the best of our knowledge, $\mathcal{L}_{\mathrm{DMI}}$ is the first loss function that is provably robust to instance-independent label noise, regardless of noise pattern, and it can be applied to any existing classification neural networks straightforwardly without any auxiliary information*. In addition to theoretical justification, we also empirically show that using $\mathcal{L}_{\mathrm{DMI}}$ outperforms all other counterparts in the classification task on both image dataset and natural language dataset include Fashion-MNIST, CIFAR-10, Dogs vs. Cats, MR with a variety of synthesized noise patterns and noise amounts, as well as a real-world dataset Clothing1M.

## 1 Introduction

Deep neural networks, together with large scale accurately annotated datasets, have achieved remarkable performance in a great many classification tasks in recent years (*e.g.*, [18, 11]). However, it is usually money- and time- consuming to find experts to annotate labels for large scale datasets. While collecting labels from crowdsourcing platforms like Amazon Mechanical Turk is a potential way to get annotations cheaper and faster, the collected labels are usually very noisy. The noisy labels hampers the performance of deep neural networks since the commonly used cross entropy loss is not noise-robust. This raises an urgent demand on designing noise-robust loss functions.

Some previous works have proposed several loss functions for training deep neural networks with noisy labels. However, they either use auxiliary information[29, 12](*e.g.*, having an additional set of clean data or the noise transition matrix) or steps[20, 33](*e.g.* estimating the noise transition matrix),

---

[*]Equal Contribution.

or make assumptions on the noise [7, 48] and thus can only handle limited kinds of the noise patterns (see perliminaries for definition of different noise patterns).

One reason that the loss functions used in previous works are not robust to a certain noise pattern, say diagonally non-dominant noise, is that they are distance-based, *i.e.*, the loss is the distance between the classifier's outputs and the labels (*e.g.* 0-1 loss, cross entropy loss). When datapoints are labeled by a careless annotator who tends to label the a priori popular class (*e.g.* For medical images, given the prior knowledge is $10\%$ malignant and $90\%$ benign, a careless annotator labels "benign" when the underline true label is "benign" and labels "benign" with 90% probability when the underline true label is "malignant".), the collected noisy labels have a diagonally non-dominant noise pattern and are extremely biased to one class ("benign"). In this situation, the distanced-based losses will prefer the "meaningless classifier" who always outputs the a priori popular class ("benign") than the classifier who outputs the true labels.

To address this issue, instead of using distance-based losses, we propose to employ information-theoretic loss such that the classifier, whose outputs have the highest mutual information with the labels, has the lowest loss. The key observation is that the "meaningless classifier" has no information about anything and will be naturally eliminated by the information-theoretic loss. Moreover, the information-monotonicity of the mutual information guarantees that adding noises to a classifier's output will make this classifier less preferred by the information-theoretic loss.

However, the key observation is not sufficient. In fact, we want an information measure I to satisfy

$$I(\text{classifier 1's output; noisy labels}) > I(\text{classifier 2's output; noisy labels})$$
$$\Leftrightarrow I(\text{classifier 1's output; clean labels}) > I(\text{classifier 2's output; clean labels}).$$

Unfortunately, the traditional Shannon mutual information (MI) does not satisfy the above formula, while we find that a generalized information measure, namely, DMI (Determinant based Mutual Information), satisfies the above formula. Like MI, DMI measures the correlation between two random variables. It is defined as the determinant of the matrix that describes the joint distribution over the two variables. Intuitively, when two random variables are independent, their joint distribution matrix has low rank and zero determinant. Moreover, DMI is not only information-monotone like MI, but also relatively invariant because of the multiplication property of the determinant. The relative invariance of DMI makes it satisfy the above formula.

Based on DMI, we propose a noise-robust loss function $\mathcal{L}_{\text{DMI}}$ which is simply

$$\mathcal{L}_{\text{DMI}}(\text{data; classifier}) := -\log[\text{DMI}(\text{classifier's output; labels})].$$

As shown in theorem 4.1 later, with $\mathcal{L}_{\text{DMI}}$, the following equation holds:

$$\mathcal{L}_{\text{DMI}}(\text{noisy data; classifier}) = \mathcal{L}_{\text{DMI}}(\text{clean data; classifier}) + \text{noise amount},$$

and the noise amount is a constant given the dataset. The equation reveals that *with $\mathcal{L}_{\text{DMI}}$, training with the noisy labels is theoretically equivalent with training with the clean labels in the dataset, regardless of the noise patterns, including the noise amounts.*

In summary, we propose a novel information theoretic noise-robust loss function $\mathcal{L}_{\text{DMI}}$ based on a generalized information measure, DMI. Theoretically we show that $\mathcal{L}_{\text{DMI}}$ is robust to instance-independent label noise. As an additional benefit, it can be easily applied to any existing classification neural networks straightforwardly without any auxiliary information. Extensive experiments have been done on both image dataset and natural language dataset including Fashion-MNIST, CIFAR-10, Dogs vs. Cats, MR with a variety of synthesized noise patterns and noise amounts as well as a real-world dataset Clothing1M. The results demonstrate the superior performance of $\mathcal{L}_{\text{DMI}}$.

## 2   Related Work

A series of works have attempted to design noise-robust loss functions. In the context of binary classification, some loss functions (*e.g.*, 0-1 loss[22], ramp loss[3], unhinged loss[40], savage loss[23]) have been proved to be robust to uniform or symmetric noise and Natarajan *et al*. [26] presented a general way to modify any given surrogate loss function. Ghosh *et al*. [7] generalized the existing results for binary classification problem to multi-class classification problem and proved that MAE (Mean Absolute Error) is robust to diagonally dominant noise. Zhang *et al*. [48] showed MAE performs poorly with deep neural network and they combined MAE and cross entropy loss to obtain

a new loss function. Patrini *et al.* [29] provided two kinds of loss correction methods with knowing the noise transition matrix. The noise transition matrix sometimes can be estimated from the noisy data [33, 20, 30]. Hendrycks *et al.* [12] proposed another loss correction technique with an additional set of clean data. To the best of our knowledge, we are the first to provide a loss function that is provably robust to instance-independent label noise without knowing the transition matrix, regardless of noise pattern and noise amount.

Instead of designing an inherently noise-robust function, several works used special architectures to deal with the problem of training deep neural networks with noisy labels. Some of them focused on estimating the noise transition matrix to handle the label noise and proposed a variety of ways to constrain the optimization [37, 43, 8, 39, 9, 44]. Some of them focused on finding ways to distinguish noisy labels from clean labels and used example re-weighting strategies to give the noisy labels less weights [31, 32, 21]. While these methods seem to perform well in practice, they cannot guarantee the robustness to label noise theoretically and are also outperformed by our method empirically.

On the other hand, Zhang *et al.* [46] have shown that deep neural networks can easily memorize completely random labels, thus several works propose frameworks to prevent this overfitting issue empirically in the setting of deep learning from noisy labels. For example, teacher-student curriculum learning framework [14] and co-teaching framework [10] have been shown to be helpful. Multi-task frameworks that jointly estimates true labels and learns to classify images are also introduced [41, 19, 38, 45]. Explicit and implicit regularization methods can also be applied [47, 25]. We consider a different perspective from them and focus on designing an inherently noise-robust function.

In this paper, we only consider instance-independent noise. There are also some works that investigate instance-dependent noise model (e.g. [5, 24]). They focus on the binary setting and assume that the noisy and true labels agree on average.

## 3  Preliminaries

### 3.1  Problem settings

We denote the set of classes by $\mathcal{C}$ and the size of $\mathcal{C}$ by $C$. We also denote the domain of datapoints by $\mathcal{X}$. A classifier is denoted by $h : \mathcal{X} \mapsto \Delta_{\mathcal{C}}$, where $\Delta_{\mathcal{C}}$ is the set of all possible distributions over $\mathcal{C}$. $h$ represents a randomized classifier such that given $x \in \mathcal{X}$, $h(x)_c$ is the probability that $h$ maps $x$ into class $c$. Note that fixing the input $x$, the randomness of a classifier is independent of everything else.

There are $N$ datapoints $\{x_i\}_{i=1}^N$. For each datapoint $x_i$, there is an *unknown* ground truth $y_i \in \mathcal{C}$. We assume that there is an unknown prior distribution $Q_{X,Y}$ over $\mathcal{X} \times \mathcal{C}$ such that $\{(x_i, y_i)\}_{i=1}^N$ are i.i.d. samples drawn from $Q_{X,Y}$ and

$$Q_{X,Y}(x, y) = \Pr[X = x, Y = y].$$

Note that here we allow the datapoints to be "imperfect" instances, *i.e.*, there still exists uncertainty for $Y$ conditioning on fully knowing $X$.

Traditional supervised learning aims to train a classifier $h^*$ that is able to classify new datapoints into their ground truth categories with access to $\{(x_i, y_i)\}_{i=1}^N$. However, in the setting of learning with noisy labels, instead, we *only* have access to $\{(x_i, \tilde{y}_i)\}_{i=1}^N$ where $\tilde{y}_i$ is a noisy version of $y_i$.

We use a random variable $\tilde{Y}$ to denote the noisy version of $Y$ and $T_{Y \to \tilde{Y}}$ to denote the transition distribution between $Y$ and , *i.e.*

$$T_{Y \to \tilde{Y}}(y, \tilde{y}) = \Pr[\tilde{Y} = \tilde{y} | Y = y].$$

We use $\mathbf{T}_{Y \to \tilde{Y}}$ to represent the $C \times C$ matrix format of $T_{Y \to \tilde{Y}}$.

Generally speaking [29, 7, 48], label noise can be divided into several kinds according to the noise transition matrix $\mathbf{T}_{Y \to \tilde{Y}}$. It is defined as *class-independent (or uniform)* if a label is substituted by a uniformly random label regardless of the classes, *i.e.* $\Pr[\tilde{Y} = \tilde{c} | Y = c] = \Pr[\tilde{Y} = \tilde{c}' | Y = c], \forall \tilde{c}, \tilde{c}' \neq c$ (e.g. $\mathbf{T}_{Y \to \tilde{Y}} = \begin{bmatrix} 0.7 & 0.3 \\ 0.3 & 0.7 \end{bmatrix}$). It is defined as *diagonally dominant* if for every row of $\mathbf{T}_{Y \to \tilde{Y}}$, the magnitude of the diagonal entry is larger than any non-diagonal entry, *i.e.* $\Pr[\tilde{Y} = c | Y = c] > \Pr[\tilde{Y} =$

$\tilde{c}|Y = c]$, $\forall \tilde{c} \neq c$ (e.g. $\mathbf{T}_{Y \to \tilde{Y}} = \begin{bmatrix} 0.7 & 0.3 \\ 0.2 & 0.8 \end{bmatrix}$). It is defined as *diagonally non-dominant* if it is not

diagonally dominant (e.g. the example mentioned in introduction, $\mathbf{T}_{Y \to \tilde{Y}} = \begin{bmatrix} 1 & 0 \\ 0.9 & 0.1 \end{bmatrix}$).

We assume that the noise is independent of the datapoints conditioning on the ground truth, which is commonly assumed in the literature [29, 7, 48], *i.e.*,

**Assumption 3.1** (Independent noise). *$X$ is independent of $\tilde{Y}$ conditioning on $Y$.*

We also need that the noisy version $\tilde{Y}$ is still informative.

**Assumption 3.2** (Informative noisy label). *$\mathbf{T}_{Y \to \tilde{Y}}$ is invertible,* i.e., $\det(\mathbf{T}_{Y \to \tilde{Y}}) \neq 0$.

### 3.2 Information theory concepts

Since Shannon's seminal work [35], information theory has shown its powerful impact in various of fields, including several recent deep learning works [13, 4, 17]. Our work is also inspired by information theory. This section introduces several basic information theory concepts.

Information theory is commonly related to random variables. For every random variable $W_1$, Shannon's entropy $\mathrm{H}(W_1) := \sum_{w_1} \Pr[W = w_1] \log \Pr[W = w_1]$ measures the uncertainty of $W_1$. For example, deterministic $W_1$ has lowest entropy. For every two random variables $W_1$ and $W_2$, Shannon mutual information $\mathrm{MI}(W_1, W_2) := \sum_{w_1, w_2} \Pr[W_1 = w_1, W_2 = w_2] \log \frac{\Pr[W=w_1, W=w_2]}{\Pr[W_1=w_1] \Pr[W_2=w_2]}$ measures the amount of relevance between $W_1$ and $W_2$. For example, when $W_1$ and $W_2$ are independent, they have the lowest Shannon mutual information, zero.

Shannon mutual information is *non-negative*, *symmetric*, *i.e.*, $\mathrm{MI}(W_1, W_2) = \mathrm{MI}(W_2, W_1)$, and also satisfies a desired property, information-monotonicity, *i.e.*, the mutual information between $W_1$ and $W_2$ will always decrease if either $W_1$ or $W_2$ has been "processed".

**Fact 3.3** (Information-monotonicity [6]). *For all random variables $W_1, W_2, W_3$, when $W_3$ is less informative for $W_2$ than $W_1$, i.e., $W_3$ is independent of $W_2$ conditioning $W_1$,*

$$MI(W_3, W_2) \leq MI(W_1, W_2).$$

This property naturally induces that for all random variables $W_1, W_2$,

$$\mathrm{MI}(W_1, W_2) \leq \mathrm{MI}(W_2, W_2) = \mathrm{H}(W_2)$$

since $W_2$ is always the most informative random variable for itself.

Based on Shannon mutual information, a performance measure for a classifier $h$ can be naturally defined. High quality classifier's output $h(X)$ should have high mutual information with the ground truth category $Y$. Thus, a classifier $h$'s performance can be measured by $\mathrm{MI}(h(X), Y)$.

However, in our setting, we only have access to the i.i.d. samples of $h(X)$ and $\tilde{Y}$. A natural attempt is to measure a classifier $h$'s performance by $\mathrm{MI}(h(X), \tilde{Y})$. Unfortunately, under this performance measure, the measurement based on noisy labels $\mathrm{MI}(h(X), \tilde{Y})$ may not be consistent with the measurement based on true labels $\mathrm{MI}(h(X), Y)$. (See a counterexample in Supplementary Material B.) That is,

$$\forall h, h', \mathrm{MI}(h(X), Y) > \mathrm{MI}(h'(X), Y) \nLeftarrow \mathrm{MI}(h(X), \tilde{Y}) > \mathrm{MI}(h'(X), \tilde{Y}).$$

Thus, we cannot use Shannon mutual information as the performance measure for classifiers. Here we find that, a generalized mutual information, Determinant based Mutual Information (DMI) [16], satisfies the above formula such that under the performance measure based on DMI, the measurement based on noisy labels is consistent with the measurement based on true labels.

**Definition 3.4** (Determinant based Mutual Information [16]). *Given two discrete random variables $W_1, W_2$, we define the Determinant based Mutual Information between $W_1$ and $W_2$ as*

$$\mathrm{DMI}(W_1, W_2) = |\det(\mathbf{Q}_{W_1, W_2})|$$

*where $\mathbf{Q}_{W_1, W_2}$ is the matrix format of the joint distribution over $W_1$ and $W_2$.*

DMI is a generalized version of Shannon's mutual information: it preserves all properties of Shannon mutual information, including non-negativity, symmetry and information-monotonicity and it is additionally relatively invariant. DMI is initially proposed to address a mechanism design problem [16].

**Lemma 3.5** (Properties of DMI [16]). *DMI is non-negative, symmetric and information-monotone. Moreover, it is relatively invariant: for all random variables $W_1, W_2, W_3$, when $W_3$ is less informative for $W_2$ than $W_1$, i.e., $W_3$ is independent of $W_2$ conditioning $W_1$,*

$$\mathrm{DMI}(W_2, W_3) = \mathrm{DMI}(W_2, W_1)|\det(\mathbf{T}_{W_1 \to W_3})|$$

*where $\mathbf{T}_{W_1 \to W_3}$ is the matrix format of*

$$T_{W_1 \to W_3}(w_1, w_3) = \Pr[W_3 = w_3 | W_1 = w_1].$$

*Proof.* The non-negativity and symmetry follow directly from the definition, so we only need to prove the relatively invariance. Note that

$$\Pr_{Q_{W_2,W_3}}[W_2 = w_2,, W_3 = w_3] = \sum_{w_1} \Pr_{Q_{W_1,W_2}}[W_2 = w_2, W_1 = w_1]\Pr[W_3 = w_3 | W_1 = w_1].$$

as $W_3$ is independent of $W_2$ conditioning on $W_1$. Thus,

$$\mathbf{Q}_{W_2,W_3} = \mathbf{Q}_{W_2,W_1}\mathbf{T}_{W_1 \to W_3}$$

where $\mathbf{Q}_{W_2,W_3}, \mathbf{Q}_{W_2,W_1}, \mathbf{T}_{W_1 \to W_3}$ are the matrix formats of $Q_{W_2,W_3}, Q_{W_2,W_1}, T_{W_1 \to W_3}$, respectively. We have

$$\det(\mathbf{Q}_{W_2,W_3}) = \det(\mathbf{Q}_{W_2,W_1})\det(\mathbf{T}_{W_1 \to W_3})$$

because of the multiplication property of the determinant (*i.e.* $\det(\mathbf{AB}) = \det(\mathbf{A})\det(\mathbf{B})$ for every two matrices $\mathbf{A}, \mathbf{B}$). Therefore, $\mathrm{DMI}(W_2, W_3) = \mathrm{DMI}(W_2, W_1)|\det(\mathbf{T}_{W_1 \to W_3})|$.

The relative invariance and the symmetry imply the information-monotonicity of DMI. When $W_3$ is less informative for $W_2$ than $W_1$, *i.e.*, $W_3$ is independent of $W_2$ conditioning on $W_1$,

$$\mathrm{DMI}(W_3, W_2) = \mathrm{DMI}(W_2, W_3) = \mathrm{DMI}(W_2, W_1)|\det(\mathbf{T}_{W_1 \to W_3})|$$
$$\leq \mathrm{DMI}(W_2, W_1) = \mathrm{DMI}(W_1, W_2)$$

because of the fact that for every square transition matrix $\mathbf{T}$, $\det(\mathbf{T}) \leq 1$ [34]. $\square$

Based on DMI, an information-theoretic performance measure for each classifier $h$ is naturally defined as $\mathrm{DMI}(h(X), \tilde{Y})$. Under this performance measure, the measurement based on noisy labels $\mathrm{DMI}(h(X), \tilde{Y})$ is consistent with the measurement based on clean labels $\mathrm{DMI}(h(X), Y)$, *i.e.*, for every two classifiers $h$ and $h'$,

$$\mathrm{DMI}(h(X), Y) > \mathrm{DMI}(h'(X), Y) \Leftrightarrow \mathrm{DMI}(h(X), \tilde{Y}) > \mathrm{DMI}(h'(X), \tilde{Y}).$$

# 4  $\mathcal{L}_{\mathrm{DMI}}$: An Information-theoretic Noise-robust Loss Function

## 4.1  Method overview

Our loss function is defined as

$$\mathcal{L}_{\mathrm{DMI}}(Q_{h(X),\tilde{Y}}) := -\log(\mathrm{DMI}(h(X), \tilde{Y})) = -\log(|\det(\mathbf{Q}_{h(X),\tilde{Y}})|)$$

where $Q_{h(X),\tilde{Y}}$ is the joint distribution over $h(X), \tilde{Y}$ and $\mathbf{Q}_{h(X),\tilde{Y}}$ is the $C \times C$ matrix format of $Q_{h(X),\tilde{Y}}$. The randomness $h(X)$ comes from both the randomness of $h$ and the randomness of $X$. The $\log$ function here resolves many scaling issues[2].

Figure 1 shows the computation of $\mathcal{L}_{\mathrm{DMI}}$. In each step of iteration, we sample a batch of datapoints and their noisy labels $\{(x_i, \tilde{y}_i)\}_{i=1}^N$. We denote the outputs of the classifier by a matrix $\mathbf{O}$. Each column of $\mathbf{O}$ is a distribution over $\mathcal{C}$, representing for an output of the classifier. We denote the noisy labels by a 0-1 matrix $\mathbf{L}$. Each row of $\mathbf{L}$ is an one-hot vector, representing for a label. i.e.

$$\mathbf{O}_{ci} = h(x_i)_c, \ \mathbf{L}_{i\tilde{c}} = \mathbb{1}[\tilde{y}_i = \tilde{c}],$$

We define $\mathbf{U} := \frac{1}{N}\mathbf{OL}$, i.e.,

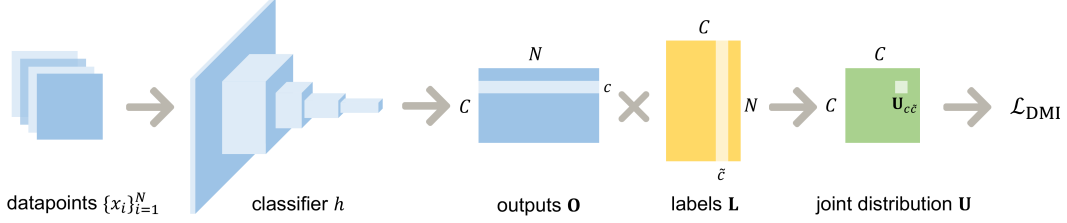

Figure 1: The computation of $\mathcal{L}_{\mathrm{DMI}}$ in each step of iteration

$$\mathbf{U}_{c\tilde{c}} := \frac{1}{N}\sum_{i=1}^{N}\mathbf{O}_{ci}\mathbf{L}_{i\tilde{c}} = \frac{1}{N}\sum_{i=1}^{N}h(x_i)_c\mathbb{1}[\tilde{y}_i = \tilde{c}].$$

We have $\mathbb{E}\mathbf{U}_{c\tilde{c}} = \Pr[h(X) = c, \tilde{Y} = \tilde{c}] = Q_{h(X),\tilde{Y}}(c,\tilde{c})$ ($\mathbb{E}$ means expectation, see proof in Supplementary Material B). Thus, $\mathbf{U}$ is an empirical estimation of $\mathbf{Q}_{h(X),\tilde{Y}}$. By abusing notation a little bit, we define

$$\mathcal{L}_{\mathrm{DMI}}(\{(x_i,\tilde{y}_i)\}_{i=1}^{N}; h) = -\log(|\det(\mathbf{U})|)$$

as the empirical loss function. Our formal training process is shown in Supplementary Material A.

## 4.2 Theoretical justification

**Theorem 4.1** (Main Theorem). *With Assumption 3.1 and Assumption 3.2, $\mathcal{L}_{\mathrm{DMI}}$ is*

**legal** *if there exists a ground truth classifier $h^*$ such that $h^*(X) = Y$, then it must have the lowest loss,* i.e., *for all classifier $h$,*

$$\mathcal{L}_{\mathrm{DMI}}(Q_{h^*(X),\tilde{Y}}) \le \mathcal{L}_{\mathrm{DMI}}(Q_{h(X),\tilde{Y}})$$

*and the inequality is strict when $h(X)$ is not a permutation of $h^*(X)$,* i.e., *there does not exist a permutation $\pi : \mathcal{C} \mapsto \mathcal{C}$ s.t. $h(x) = \pi(h^*(x)), \forall x \in \mathcal{X}$;*

**noise-robust** *for the set of all possible classifiers $\mathcal{H}$,*

$$\arg\min_{h\in\mathcal{H}} \mathcal{L}_{\mathrm{DMI}}(Q_{h(X),\tilde{Y}}) = \arg\min_{h\in\mathcal{H}} \mathcal{L}_{\mathrm{DMI}}(Q_{h(X),Y})$$

*and in fact, training using noisy labels is the same as training using clean labels in the dataset except a constant shift,*

$$\mathcal{L}_{\mathrm{DMI}}(Q_{h(X),\tilde{Y}}) = \mathcal{L}_{\mathrm{DMI}}(Q_{h(X),Y}) + \alpha;$$

**information-monotone** *for every two classifiers $h, h'$, if $h'(X)$ is less informative for $Y$ than $h(X)$, i.e. $h'(X)$ is independent of $Y$ conditioning on $h(X)$, then*

$$\mathcal{L}_{\mathrm{DMI}}(Q_{h'(X),\tilde{Y}}) \le \mathcal{L}_{\mathrm{DMI}}(Q_{h(X),\tilde{Y}}).$$

*Proof.* The relatively invariance of DMI (Lemma 3.5) implies

$$\mathrm{DMI}(h(X), \tilde{Y}) = \mathrm{DMI}(h(X), Y)|\det(\mathbf{T}_{Y\to\tilde{Y}})|.$$

Therefore,

$$\mathcal{L}_{\mathrm{DMI}}(Q_{h(X),\tilde{Y}}) = \mathcal{L}_{\mathrm{DMI}}(Q_{h(X),Y}) + \log(|\det(\mathbf{T}_{Y\to\tilde{Y}})|).$$

Thus, the information-monotonicity and the noise-robustness of $\mathcal{L}_{\mathrm{DMI}}$ follows and the constant $\alpha = \log(|\det(\mathbf{T}_{Y\to\tilde{Y}})|) \le 0$.

The legal property follows from the information-monotonicity of $\mathcal{L}_{\mathrm{DMI}}$ as $h^*(X) = Y$ is the most informative random variable for $Y$ itself and the fact that for every square transition matrix $T$, $\det(T) = 1$ if and only if $T$ is a permutation matrix [34]. □

# 5 Experiments

We evaluate our method on both synthesized and real-world noisy datasets with different deep neural networks to demonstrate that our method is independent of both architecture and data domain. We call our method **DMI** and compare it with: **CE** (the cross entropy loss), **FW** (the forward loss [29]), **GCE** (the generalized cross entropy loss [48]), **LCCN** (the latent class-conditional noise model [44]). For the synthesized data, noises are added to the training and validation sets, and test accuracy is computed with respect to true labels. For our method, we pick the best learning rate from $\{1.0 \times 10^{-4}, 1.0 \times 10^{-5}, 1.0 \times 10^{-6}\}$ and the best batch size from $\{128, 256\}$ based on the minimum validation loss. For other methods, we use the best hyperparameters they provided in similar settings. The classifiers are pretrained with cross entropy loss first. All reported experiments were repeated five times. We implement all networks and training procedures in Pytorch [28] and conduct all experiments on NVIDIA TITAN Xp GPUs.[3] The explicit noise transition matrices are shown in Supplementary Material C. Due to space limit, we defer some additional experiments to Supplementary Material D.

## 5.1 An explanation experiment on Fashion-MNIST

To compare distance-based and information-theoretic loss functions as we mentioned in the third paragraph in introduction, we conducted experiments on Fashion-MNIST [42]. It consists of 70,000 $28 \times 28$ grayscale fashion product image from 10 classes, which is split into a $50,000$-image training set, a $10,000$-image valiadation set and a $10,000$-image test set. For clean presentation, we only compare our information-theoretic loss function **DMI** with the distance-based loss function **CE** here and convert the labels in the dataset to two classes, bags and clothes, to synthesize a highly imbalanced dataset ($10\%$ bags, $90\%$ clothes). We use a simple two-layer convolutional neural network as the classifier. Adam with default parameters and a learning rate of $1.0 \times 10^{-4}$ is used as the optimizer during training. Batch size is set to $128$.

We synthesize three cases of noise patterns: (1) with probability $r$, a true label is substituted by a random label through uniform sampling. (2) with probability $r$, bags $\rightarrow$ clothes, that is, a true label of the a priori less popular class, "bags", is flipped to the popular one, "clothes". This happens in real world when the annotators are lazy. (*e.g.*, a careless medical image annotator may be more likely to label "benign" since most images are in the "benign" category.) (3) with probability $r$, clothes $\rightarrow$ bags, that is, the a priori more popular class, "clothes", is flipped to the other one, "bags". This happens in real world when the annotators are risk-avoid and there will be smaller adverse effects if the annotators label the image to a certain class. (*e.g.* a risk-avoid medical image annotator may be more likely to label "malignant" since it is usually safer when the annotator is not confident, even if it is less likely a priori.) Note that the parameter $0 \le r \le 1$ in the above three cases also represents the amount of noise. When $r = 0$, the labels are clean and when $r = 1$, the labels are totally uninformative. Moreover, in case (2) and (3), as $r$ increases, the noise pattern changes from diagonally dominant to diagonally non-dominant.

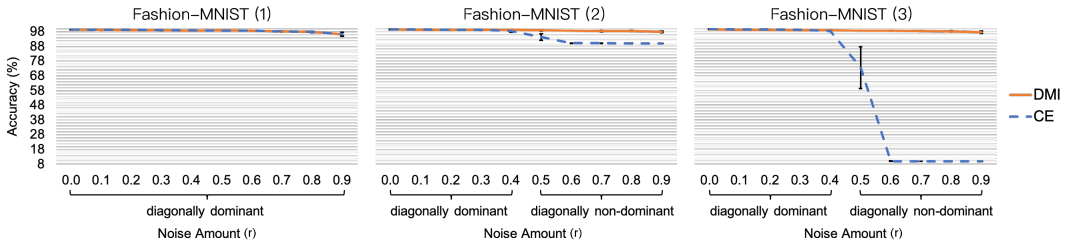

Figure 2: Test accuracy (mean and std. dev.) on Fashion-MNIST.

As we mentioned in the introduction, distance-based loss functions will perform badly when the noise is non-diagonally dominant and the labels are biased to one class since they prefer the meaningless classifier $h_0$ who always outputs the class who is the majority in the labels. ($\forall x, h_0(x) =$ "clothes" and has accuracy $90\%$ in case (2) and $\forall x, h_0(x) =$ "bags" and has accuracy $10\%$ in case (3)). The

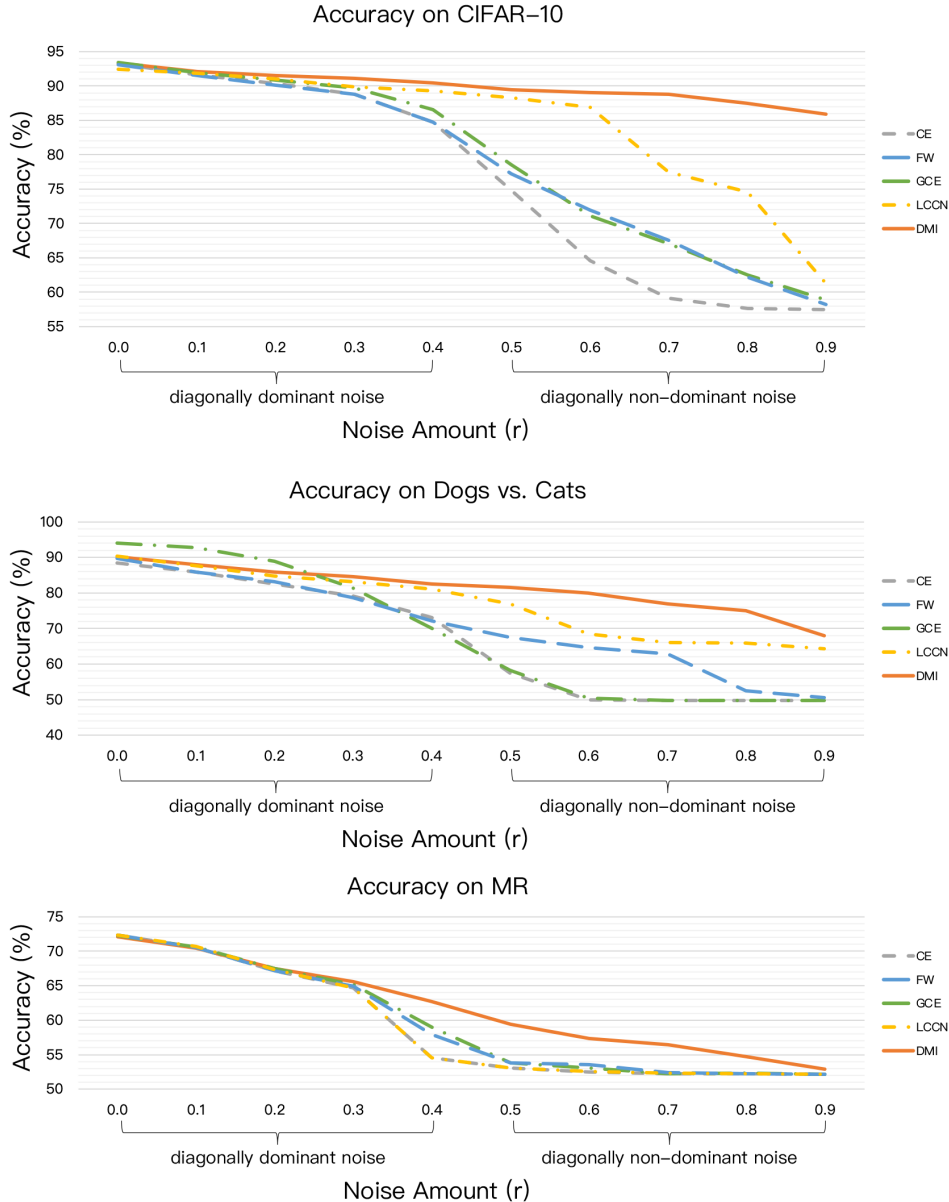

Figure 3: Test accuracy (mean) on CIFAR-10, Dogs vs. Cats and MR.

experiment results match our expectation. **CE** performs similarly with our **DMI** for diagonally dominant noises. For non-diagonally dominant noises, however, **CE** only obtains the meaningless classifier $h_0$ while **DMI** still performs pretty well.

## 5.2 Experiments on CIFAR-10, Dogs vs. Cats and MR

CIFAR-10 [1] consists of 60,000 $32 \times 32$ color images from 10 classes, which is split into a $40,000$-image training set, a $10,000$-image validation set and a $10,000$-image test set. Dogs vs. Cats [2] consists of $25,000$ images from 2 classes, dogs and cats, which is split into a $12,500$-image training set, a $6,250$-image validation set and a $6,250$-image test set. MR [27] consist of $10,662$ one-sentence movie reviews from 2 classes, positive and negative, which is split into a $7,676$-sentence training set, a $1,919$-sentence validation set and a $1,067$-sentence test set. We use ResNet-34[11], VGG-16[36], WordCNN[15] as the classifier for CIFAR-10, Dogs vs. Cats, MR, respectively. SGD with a momentum of $0.9$, a weight decay of $1.0 \times 10^{-4}$ and a learning rate of $1.0 \times 10^{-5}$ is used as the

optimizer during training for CIFAR-10 and Dogs vs. Cats. Adam with default parameters and a learning rate of $1.0 \times 10^{-4}$ is used as the optimizer during training for MR. Batch size is set to 128. We use per-pixel normalization, horizontal random flip and $32 \times 32$ random crops after padding with 4 pixels on each side as data augmentation for images in CIFAR-10 and Dogs vs Cats. We use the same pre-processing pipeline in [15] for sentences in MR. Following [44], the noise for CIFAR-10 is added between the similar classes, i.e. truck → automobile, bird → airplane, deer → horse, cat → dog, with probability $r$. The noise for Dogs vs. Cats is added as cat → dog with probability $r$. The noise for MR is added as positive → negative with probability $r$.

As shown in Figure 3, our method **DMI** almost outperforms all other methods in every experiment and its accuracy drops slowly as the noise amount increases. **GCE** has great performance in diagonally dominant noises but it fails in diagonally non-dominant noises. This phenomenon matches its theory: it assumes that the label noise is diagonally dominant. **FW** needs to pre-estimate a noise transition matrix before training and **LCCN** uses the output of the model to estimate the true labels. These tasks become harder as the noise amount grows larger, so their performance also drop quickly as the noise amount increases.

## 5.3 Experiments on Clothing1M

Clothing1M [43] is a large-scale real world dataset, which consists of 1 million images of clothes collected from shopping websites with noisy labels from 14 classes assigned by the surrounding text provided by the sellers. It has additional 14k and 10k clean data respectively for validation and test. We use ResNet-50[11] as the classifier and apply random crop of $224 \times 224$, random flip, brightness and saturation as data augmentation. SGD with a momentum of 0.9, a weight decay of $1.0 \times 10^{-3}$ is used as the optimizer during training. We train the classifier with learning rates of $1.0 \times 10^{-6}$ in the first 5 epochs and $0.5 \times 10^{-6}$ in the second 5 epochs. Batch size is set to 256.

Table 1: Test accuracy (mean) on Clothing1M

| Method | CE | FW | GCE | LCCN | DMI |
|---|---|---|---|---|---|
| Accuracy | 68.94 | 70.83 | 69.09 | 71.63 | **72.46** |

As shown in Table 5, **DMI** also outperforms other methods in the real-world setting.

# 6 Conclusion and Discussion

We propose a simple yet powerful loss function, $\mathcal{L}_{\mathrm{DMI}}$, for training deep neural networks robust to label noise. It is based on a generalized version of mutual information, DMI. We provide theoretical validation to our approach and compare our approach experimentally with previous methods on both synthesized and real-world datasets. To the best of our knowledge, $\mathcal{L}_{\mathrm{DMI}}$ is the first loss function that is provably robust to instance-independent label noise, regardless of noise pattern and noise amount, and it can be applied to any existing classification neural networks straightforwardly without any auxiliary information.

In the experiment, sometimes **DMI** does not have advantage when the data is clean and is outperformed by **GCE**. **GCE** does a training optimization on MAE with some hyperparameters while sacrifices the robustness a little bit theoretically. A possible future direction is to employ some training optimizations in our method to improve the performance.

The current paper focuses on the instance-independent noise setting. That is, we assume conditioning on the latent ground truth label $Y$, $\tilde{Y}$ and $X$ are independent. There may exist $Y' \neq Y$ such that $\tilde{Y}$ and $X$ are independent conditioning on $Y'$. Based on our theorem, training using $\tilde{Y}$ is also the same as training using $Y'$. However, without any additional assumption, when we only has the conditional independent assumption, no algorithm can distinguish $Y'$ and $Y$. Moreover, the information-monotonicity of our loss function guarantees that if $Y$ is more informative than $Y'$ with $X$, the best hypothesis learned in our algorithm will be more similar with $Y$ than $Y'$. Thus, if we assume that the actual ground truth label $Y$ is the most informative one, then our algorithm can learn to predict $Y$ rather than other $Y'$s. An interesting future direction is to combine our method with additional assumptions to give a better prediction.

**Acknowledgments**

We would like to express our thanks for support from the following research grants: 2018AAA0102004, NSFC-61625201, NSFC-61527804.

## Footnotes

[2] $\frac{\partial(c|\det(\mathbf{A})|)}{\partial \mathbf{A}} = c|\det(\mathbf{A})|(\mathbf{A}^{-1})^T$ while $\frac{\partial \log(c|\det(\mathbf{A})|)}{\partial \mathbf{A}} = (\mathbf{A}^{-1})^T$, $\forall$ matrix $\mathbf{A}$ and $\forall$ constant $c$.

[3]Source codes are available at `https://github.com/Newbeeer/L_DMI`.

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
