[Supplementary Material]

# A  Training Process

---
**Algorithm 1** The training process with $\mathcal{L}_{\mathrm{DMI}}$

---
**Require:** A training dataset $\mathcal{D} = \{(x_i, \tilde{y}_i)\}_{i=1}^{D}$, a validation dataset $\mathcal{V} = \{(x_i, \tilde{y}_i)\}_{i=1}^{V}$, a classifier modeled by deep neural network $h_\Theta$, the running epoch number $T$, the learning rate $\gamma$ and the batch size $N$.
1: Pretrain the classifier $h_\Theta$ on the dataset $\mathcal{D}$ with cross entropy loss
2: Initialize the best classifier: $h_{\Theta^*} \leftarrow h_\Theta$
3: Randomly sample a batch of samples $\mathcal{B}_v = \{(x_i, \tilde{y}_i)\}_{i=1}^{N}$ from the validation dataset
4: Initialize the minimum validation loss: $L^* \leftarrow \mathcal{L}_{\mathrm{DMI}}(\mathcal{B}_v; h_\Theta)$
5: **for** epoch $t = 1 \to T$ **do**
6:   **for** batch $b = 1 \to \lceil D/B \rceil$ **do**
7:     Randomly sample a batch of samples $\mathcal{B}_t = \{(x_i, \tilde{y}_i)\}_{i=1}^{N}$ from the training dataset
8:     Compute the training loss: $L \leftarrow \mathcal{L}_{\mathrm{DMI}}(\mathcal{B}_t; h_\Theta)$
9:     Update $\Theta$: $\Theta \leftarrow \Theta - \gamma \frac{\partial L}{\partial \Theta}$
10:   **end for**
11:   Randomly sample a batch of samples $\mathcal{B}_v = \{(x_i, \tilde{y}_i)\}_{i=1}^{N}$ from the validation dataset
12:   Compute the validation loss: $L \leftarrow \mathcal{L}_{\mathrm{DMI}}(\mathcal{B}_v; h_\Theta)$
13:   **if** $L < L^*$ **then**
14:     Update the minimum validation loss: $L^* \leftarrow L$
15:     Update the best classifier: $h_{\Theta^*} \leftarrow h_\Theta$
16:   **end if**
17: **end for**
18: **return** the best classifier $h_{\Theta^*}$

---

# B  Other Proofs

**Claim B.1.**
$$\mathbb{E}\mathbf{U}_{c\tilde{c}} = \Pr[h(X) = c, \tilde{Y} = \tilde{c}]$$
*where*
$$\mathbf{U}_{c\tilde{c}} := \frac{1}{N}\sum_{i=1}^{N} \mathbf{O}_{ci}\mathbf{L}_{i\tilde{c}} = \frac{1}{N}\sum_{i=1}^{N} h(x_i)_c \mathbb{1}[\tilde{y}_i = \tilde{c}].$$

*Proof.* Recall that the randomness of $h(X)$ comes from both $h$ and $X$ and the randomness of $h$ is independent of everything else.

$$
\begin{aligned}
\mathbb{E}\mathbf{U}_{c\tilde{c}} &= \mathbb{E}\frac{1}{N}\sum_{i=1}^{N} h(x_i)_c \mathbb{1}[\tilde{y}_i = \tilde{c}] \\
&= \mathbb{E}_{X,\tilde{Y}} h(X)_c \mathbb{1}[\tilde{Y} = \tilde{c}] && \text{(i.i.d. samples)} \\
&= \sum_{x,\tilde{y}} \Pr[X = x, \tilde{Y} = \tilde{y}] h(x)_c \mathbb{1}[\tilde{y} = \tilde{c}] \\
&= \sum_{x} \Pr[X = x, \tilde{Y} = \tilde{c}] h(x)_c \\
&= \sum_{x} \Pr[X = x, \tilde{Y} = \tilde{c}] \Pr[h(X) = c | X = x] && \text{(definition of randomized classifier)} \\
&= \sum_{x} \Pr[X = x, \tilde{Y} = \tilde{c}] \Pr[h(X) = c | X = x, \tilde{Y} = \tilde{c}] \\
&&& \text{(fixing } x \text{, the randomness of } h \text{ is independent of everything else)} \\
&= \Pr[h(X) = c, \tilde{Y} = \tilde{c}].
\end{aligned}
$$

$\square$

**Claim B.2.** *Under the the performance measure based on Shannon mutual information, the measurement based on noisy labels $MI(h(X),\tilde{Y})$ is not consistent with the measurement based on true labels $MI(h(X),Y)$. i.e., for every two classifiers $h$ and $h'$,*

$$I(h(X),Y) > I(h'(X),Y) \not\Rightarrow I(h(X),\tilde{Y}) > I(h'(X),\tilde{Y}).$$

*Proof.* See a counterexample:

The matrix format of the joint distribution $Q_{h(X),Y}$ is $\mathbf{Q}_{h(X),Y} = \begin{bmatrix} 0.1 & 0.4 \\ 0.2 & 0.3 \end{bmatrix}$, the matrix format of the joint distribution $Q_{h'(X),Y}$ is $\mathbf{Q}_{h'(X),Y} = \begin{bmatrix} 0.2 & 0.6 \\ 0.1 & 0.1 \end{bmatrix}$ and the noise transition matrix is $\mathbf{T}_{Y \to \tilde{Y}} = \begin{bmatrix} 0.8 & 0.2 \\ 0.4 & 0.6 \end{bmatrix}$.

Given these conditions, $\mathbf{Q}_{h(X),\tilde{Y}} = \begin{bmatrix} 0.24 & 0.26 \\ 0.28 & 0.22 \end{bmatrix}$ and $\mathbf{Q}_{h'(X),\tilde{Y}} = \begin{bmatrix} 0.40 & 0.40 \\ 0.12 & 0.08 \end{bmatrix}$.

If we use Shannon mutual information as the performance measure,

$$MI(h(X),Y) = 2.4157 \times 10^{-2},$$
$$MI(h'(X),Y) = 2.2367 \times 10^{-2},$$
$$MI(h(X),\tilde{Y}) = 3.2085 \times 10^{-3},$$
$$MI(h'(X),\tilde{Y}) = 3.2268 \times 10^{-3}.$$

Thus we have $MI(h(X),Y) > MI(h'(X),Y)$ but $MI(h(X),\tilde{Y}) < MI(h'(X),\tilde{Y})$.

Therefore, $MI(h(X),Y) > MI(h'(X),Y) \not\Rightarrow MI(h(X),\tilde{Y}) > MI(h'(X),\tilde{Y})$.

$\square$

## C  Noise Transition Matrices

Here we list the explicit noise transition matrices.

On Fashion-MNIST, case (1): $\mathbf{T}_{Y \to \tilde{Y}} = \begin{bmatrix} 1 - \frac{r}{2} & \frac{r}{2} \\ \frac{r}{2} & 1 - \frac{r}{2} \end{bmatrix}$;

On Fashion-MNIST, case (2): $\mathbf{T}_{Y \to \tilde{Y}} = \begin{bmatrix} 1 - r & r \\ 0 & 1 \end{bmatrix}$;

On Fashion-MNIST, case (3): $\mathbf{T}_{Y \to \tilde{Y}} = \begin{bmatrix} 1 & 0 \\ r & 1 - r \end{bmatrix}$;

On CIFAR-10, $\mathbf{T}_{Y \to \tilde{Y}} = \begin{bmatrix} 1 & 0 & 0 & 0 & 0 & 0 & 0 & 0 & 0 & 0 \\ 0 & 1 & 0 & 0 & 0 & 0 & 0 & 0 & 0 & 0 \\ r & 0 & 1-r & 0 & 0 & 0 & 0 & 0 & 0 & 0 \\ 0 & 0 & 0 & 1-r & 0 & r & 0 & 0 & 0 & 0 \\ 0 & 0 & 0 & 0 & 1-r & 0 & 0 & r & 0 & 0 \\ 0 & 0 & 0 & 0 & 0 & 1 & 0 & 0 & 0 & 0 \\ 0 & 0 & 0 & 0 & 0 & 0 & 1 & 0 & 0 & 0 \\ 0 & 0 & 0 & 0 & 0 & 0 & 0 & 1 & 0 & 0 \\ 0 & 0 & 0 & 0 & 0 & 0 & 0 & 0 & 1 & 0 \\ 0 & r & 0 & 0 & 0 & 0 & 0 & 0 & 0 & 1-r \end{bmatrix}$;

On Dogs vs. Cats, $\mathbf{T}_{Y \to \tilde{Y}} = \begin{bmatrix} 1 & 0 \\ r & 1 - r \end{bmatrix}$.

On MR, $\mathbf{T}_{Y \to \tilde{Y}} = \begin{bmatrix} 1 & 0 \\ r & 1 - r \end{bmatrix}$.

For Fashion-MNIST case (1), $r = 0.0, 0.1, 0.2, 0.3, 0.4, 0.5, 0.6, 0.7, 0.8, 0, 9$ are diagonally dominant noises. For other cases, $r = 0.0, 0.1, 0.2, 0.3, 0.4$ are diagonally dominant noises and $r = 0.5, 0.6, 0.7, 0.8, 0, 9$ are diagonally non-dominant noises.

# D  Additional Experiments

For clean presentation, we only include the comparison between **CE** and **DMI** in section 5.1 and attach comparisons with other methods here. In the experiments in section 5.2, noise patterns are divided into two main cases, diagonally dominant and diagonally non-dominant and uniform noise is a special case of diagonally dominant noise. Thus, we did not emphasize the uniform noise results in section 5.2 and attach them here.

Figure 4: Additional experiments

We also compared our method to **MentorNet** (the sample reweighting loss [14]) and **VAT** (the regularization loss [25]). For clean presentation, we only attach them here. Our method still outperforms these two additional baselines in most of the cases. [4]

Table 2: Test accuracy on CIFAR-10 (mean ± std. dev.)

| $r$ | CE | MentorNet | VAT | FW | GCE | LCCN | DMI |
|---|---|---|---|---|---|---|---|
| 0.0 | $93.29 \pm 0.18$ | $92.13 \pm 1.22$ | $92.25 \pm 0.1$ | $93.12 \pm 0.16$ | $\mathbf{93.43 \pm 0.24}$ | $92.47 \pm 0.36$ | $93.37 \pm 0.20$ |
| 0.1 | $91.63 \pm 0.32$ | $91.35 \pm 0.83$ | $91.4 \pm 0.68$ | $91.54 \pm 0.15$ | $91.96 \pm 0.09$ | $91.88 \pm 0.23$ | $\mathbf{92.08 \pm 0.08}$ |
| 0.2 | $90.36 \pm 0.24$ | $90.06 \pm 0.52$ | $91.19 \pm 0.31$ | $90.10 \pm 0.22$ | $90.87 \pm 0.16$ | $91.05 \pm 0.43$ | $\mathbf{91.51 \pm 0.17}$ |
| 0.3 | $88.79 \pm 0.40$ | $88.47 \pm 0.61$ | $88.97 \pm 0.41$ | $88.77 \pm 0.36$ | $89.67 \pm 0.21$ | $89.88 \pm 0.40$ | $\mathbf{91.12 \pm 0.30}$ |
| 0.4 | $84.76 \pm 0.98$ | $84.12 \pm 1.29$ | $84.09 \pm 0.46$ | $84.78 \pm 1.53$ | $86.6 \pm 0.47$ | $89.33 \pm 0.58$ | $\mathbf{90.41 \pm 0.32}$ |
| 0.5 | $74.81 \pm 3.37$ | $78.43 \pm 0.39$ | $75.07 \pm 0.66$ | $77.2 \pm 4.19$ | $78.53 \pm 1.93$ | $88.30 \pm 0.38$ | $\mathbf{89.45 \pm 0.99}$ |
| 0.6 | $64.61 \pm 0.72$ | $71.33 \pm 0.13$ | $65.02 \pm 0.63$ | $71.98 \pm 1.83$ | $71.14 \pm 0.78$ | $86.89 \pm 0.51$ | $\mathbf{89.03 \pm 0.69}$ |
| 0.7 | $59.15 \pm 0.64$ | $66.28 \pm 0.76$ | $58.92 \pm 1.49$ | $67.59 \pm 1.64$ | $67.10 \pm 0.82$ | $77.50 \pm 0.60$ | $\mathbf{88.82 \pm 0.89}$ |
| 0.8 | $57.65 \pm 0.28$ | $65.67 \pm 0.57$ | $57.78 \pm 0.32$ | $62.22 \pm 1.80$ | $62.56 \pm 0.72$ | $74.62 \pm 1.16$ | $\mathbf{87.46 \pm 0.79}$ |
| 0.9 | $57.46 \pm 0.08$ | $59.49 \pm 0.40$ | $57.19 \pm 1.25$ | $58.23 \pm 0.25$ | $58.91 \pm 0.46$ | $61.32 \pm 1.87$ | $\mathbf{85.94 \pm 0.74}$ |

Table 3: Test accuracy on Dogs vs. Cats (mean ± std. dev.)

| $r$ | CE | MentorNet | VAT | FW | GCE | LCCN | DMI |
|---|---|---|---|---|---|---|---|
| 0.0 | 88.50 ± 0.60 | 88.76 ± 0.32 | 88.32 ± 0.76 | 89.66 ± 0.63 | **94.06 ± 0.41** | 90.41 ± 0.38 | 90.21 ± 0.27 |
| 0.1 | 85.87 ± 0.79 | 87.33 ± 0.51 | 87.04 ± 1.53 | 85.87 ± 0.54 | **92.75 ± 0.50** | 87.72 ± 0.46 | 87.99 ± 0.41 |
| 0.2 | 82.50 ± 0.96 | 82.08 ± 0.60 | 82.36 ± 0.78 | 83.20 ± 0.83 | **88.94 ± 0.70** | 84.80 ± 0.93 | 85.88 ± 0.83 |
| 0.3 | 79.11 ± 1.08 | 80.14 ± 0.99 | 78.55 ± 0.76 | 78.71 ± 1.97 | 81.34 ± 3.23 | 83.16 ± 1.18 | **84.61 ± 0.98** |
| 0.4 | 73.05 ± 0.20 | 72.24 ± 0.75 | 74.72 ± 0.57 | 72.13 ± 2.42 | 70.13 ± 3.59 | 81.06 ± 1.05 | **82.52 ± 1.01** |
| 0.5 | 57.46 ± 3.71 | 63.62 ± 0.39 | 66.83 ± 0.75 | 67.50 ± 3.99 | 58.31 ± 1.19 | 76.88 ± 2.97 | **81.50 ± 1.19** |
| 0.6 | 49.98 ± 0.15 | 63.07 ± 0.93 | 55.02 ± 1.41 | 64.58 ± 5.21 | 50.39 ± 0.47 | 68.50 ± 3.40 | **80.00 ± 0.72** |
| 0.7 | 49.83 ± 0.09 | 52.38 ± 0.66 | 54.18 ± 0.72 | 62.87 ± 6.82 | 49.76 ± 0.00 | 66.10 ± 2.45 | **77.01 ± 1.07** |
| 0.8 | 49.80 ± 0.03 | 51.42 ± 0.75 | 51.88 ± 0.25 | 52.44 ± 1.52 | 49.76 ± 0.00 | 65.93 ± 2.76 | **75.01 ± 0.88** |
| 0.9 | 49.77 ± 0.01 | 51.31 ± 0.20 | 51.69 ± 0.70 | 50.56 ± 1.32 | 49.76 ± 0.00 | 64.29 ± 1.46 | **67.96 ± 1.45** |

Table 4: Test accuracy on MR (mean ± std. dev.)

| $r$ | CE | MentorNet | FW | GCE | LCCN | DMI |
|---|---|---|---|---|---|---|
| 0.0 | 72.35 ± 0.00 | **72.44 ± 0.32** | 72.35 ± 0.00 | 72.24 ± 0.10 | 72.35 ± 0.00 | 72.07 ± 0.00 |
| 0.1 | 70.51 ± 0.97 | 69.54 ± 0.19 | 70.49 ± 0.94 | **70.58 ± 1.03** | 70.72 ± 1.02 | 70.42 ± 0.73 |
| 0.2 | 67.12 ± 1.19 | 66.72 ± 0.98 | 67.14 ± 1.21 | **67.48 ± 1.02** | 67.33 ± 1.61 | 67.44 ± 1.22 |
| 0.3 | 64.68 ± 1.22 | 65.13 ± 0.13 | 64.92 ± 1.37 | 65.19 ± 1.09 | 64.65 ± 1.58 | **65.62 ± 1.04** |
| 0.4 | 54.52 ± 1.74 | 54.73 ± 1.01 | 57.89 ± 2.51 | 58.97 ± 1.77 | 54.52 ± 1.74 | **62.67 ± 2.27** |
| 0.5 | 53.08 ± 0.64 | 53.70 ± 0.55 | 53.83 ± 0.68 | 53.81 ± 2.04 | 53.08 ± 0.64 | **59.40 ± 0.63** |
| 0.6 | 52.52 ± 0.57 | 53.15 ± 0.97 | 53.58 ± 0.35 | 53.08 ± 1.46 | 52.54 ± 0.59 | **57.38 ± 0.81** |
| 0.7 | 52.28 ± 0.12 | 52.76 ± 0.98 | 52.38 ± 0.19 | 52.22 ± 0.10 | 52.29 ± 0.13 | **56.44 ± 0.78** |
| 0.8 | 52.26 ± 0.08 | 52.29 ± 0.25 | 52.24 ± 0.08 | 52.31 ± 0.15 | 52.25 ± 0.08 | **54.69 ± 0.65** |
| 0.9 | 52.20 ± 0.00 | 52.20 ± 0.56 | 52.16 ± 0.14 | 52.20 ± 0.07 | 52.20 ± 0.00 | **52.88 ± 0.33** |

Table 5: Test accuracy (mean) on Clothing1M

| Method | CE | MentorNet | VAT | FW | GCE | LCCN | DMI |
|---|---|---|---|---|---|---|---|
| Accuracy | 68.94 | 69.30 | 69.57 | 70.83 | 69.09 | 71.63 | **72.46** |

## Footnotes

[4]**VAT** can not be applied to MR dataset.