[Reviews · NeurIPS 2019]

Reviewer 1



Overall, the paper is well organized and easy to follow. The problem of learning dnn robust to label noise is interesting in both research and practical application. The proposed robust loss function sounds reasonable and should be effectivess in some case. For the weakness of the paper, see the following improvements suggestion.

Reviewer 2



Label noise learning is a hot topic now as the datasets grow bigger and the labels are becoming noisier. How to learn the optimal classifier w.r.t. the clean data from the noisy data is challenging. To guarantee to learn the optimal classifier, many robust learning methods have been proposed. To the best of my knowledge, they all need the information of the transition matrix, learning which could be challenging. This paper proposes the first loss function that is robust to instance-independent label noise without knowing the transition matrix. Thus, making a significant contribution to the community. My main concern is that once the latent true variable Y is not identifiable, e.g., there exists another latent variable Y', which also has a transition relationship with the noisy label. How to find the optimal classifier for Y? This case generally exists when the transition matrix is not identifiable. From the experiments, we find that the proposed method doesn't perform well when the noise rate is low. Intuitively, the proposed method should work well for clean data as well. The authors are suggested to do some comparisons on clean data and explain why it doesn't well well on small noise. === After seeing the rebuttal, my main concern hasn’t been well addressed. The current paper has an issue that it is unclear if the latent true label Y is identifiable in the proposed method. It seems that if there is a latent variable Y’ which also has a constant transition matrix with the noisy labels. The proposed method cannot distinguish Y and Y’. The authors response that the proposed method will learn something more informative about X. This is reasonable, but it also implies that if Y’ is more informative about X. The proposed method will lead to wrong predictions. For example, in the case that instances potentially have multiple classes and there are constant transition relationships among them, it may be very hard for the proposed method to distinguish them. It seems necessary to illustrate if the proposed method works well on clean data. If it did not work well on the clean data, it may be caused by the reason that the proposed method finds a Y’ instead of Y. I am worrying that the authors avoid learning the transition matrix but were introducing a harder problem of identifying Y.

Reviewer 3



This paper proposes a new information theoretic loss function, L_DML, for training deep neural networks robust to noisy labels. Specifically, this paper first proposes a new information measure, DMI (Determinant based Mutual Information), which is a generalized version of mutual information. Based on the relative invariance of DMI, this paper proposes a noise-robust loss function called L_DMI, which is theoretically justified. Experiments on synthetic and real-world noisy datasets demonstrate the effectiveness of the proposed L_DMI on defending diagonally dominant and diagonally non-dominant noise. Pros: 1) The proposed information measure DMI, and robust loss function L_DMI, are both theoretically justified. 2) The proposed L_DMI loss function is easy to implement. 3) Empirical results on synthetic and real-world noisy datasets show that L_DMI outperforms other baselines. Cons: 1) The results on Fashion-MNIST show that the proposed method is not sensitive to noise patterns (i.e. class-independent and class-dependent noises), and noise amount (with probability from 0.1 to 0.9). However, it is unclear why converting Fashion-MNIST to two classes instead of using the original 10-class setting. Can explain more about that? To maintain consistency, the comparison with more baselines (i.e. LCCN, GCE, and FW) should also be provided. 2) For CIFAR-10 dataset, the proposed method is only evaluated by adding noise to similar classes. The evaluation on the uniform noise is missing. 3) For Dogs vs. Cats, why the experiment setting is not consistent with Fashion-MNIST, which is also two-class case? Specifically, the evaluations on uniform noise and ‘dog->cat’ noise are missing. Minor issue Line 283: “neutral networks” -> “neural networks” Overall, this paper proposes a new noise-robust loss function for defending label noise. The proposed method is both theoretically and empirically sound.

Reviewer 4



LDMI: A Novel Information-theoretic Loss Function for Training Deep Nets Robust to Label Noise: This paper formulates a new information-theoretic loss function, which is based on determinant based mutual information (DMI). Their main contribution is that this loss is provably not senstive to noise patterns and noise amounts. Pros: 1. The authors find a relatively new direction for learning with noisy labels. Namely, instead of designing distance-based losses, they try the information-theoretical loss. Based on this motivation, they design DMI loss, which is robust to label noise. 2. Related works: In deep learning with noisy labels, there are several main directions, including robust loss functions [1], reweighting trick [2], and explicit and implicit regularization [3]. I indeed appreciate authors survey them well. Note that, the authors may cite [4] in the regularization line due to its high impact. 3. The authors perform numerical experiments to demonstrate the efficacy of their framework. And their experimental result support their previous claims. For example, they conduct experiments on Fashion-MNIST, CIFAR-10 and Dogs vs. Cats. Besides, they conduct experiments on Clothing1M dataset [5]. Cons: We have two questions in the following. 1. Settings: This paper still focuses on class-conditional noise (CCN) model. However, CCN model may not cover the real-world noise case. The current emerging noise model is instance-dependent noise model [6,7]. I am not sure whether this idea can be depolyed under this case. 2. Experiments: 2.1 Datasets: I think the author should conduct 1 NLP dataset instead of only using image datasets. 2.2 Baselines: Please add the results from reweighting methods like MentorNet [2]; Please compare your method with VAT [4]. References: [1] Z. Zhang and M. Sabuncu. Generalized cross entropy loss for training deep neural networks with noisy labels. In NeurIPS, 2018. [2] L. Jiang, Z. Zhou, T. Leung, L. Li, and L. Fei-Fei. Mentornet: Learning data-driven curriculum for very deep neural networks on corrupted labels. In ICML, 2018. [3] H. Zhang, M. Cisse, Y.N. Dauphin, and Y.N. Lopez-Paz. Mixup: Beyond empirical risk minimization. In ICLR, 2018. [4] T. Miyato, S. Maeda, M. Koyama, and S. Ishii. Virtual adversarial training: A regularization method for supervised and semi-supervised learning. In ICLR, 2016. [5] T. Xiao, T. Xia, Y. Yang, C. Huang, and X. Wang. Learning from massive noisy labeled data for image classification. In CVPR, 2015. [6] J. Cheng, T. Liu, K. Rao, and D. Tao. Learning with bounded instance-and label-dependent label noise. arXiv 1709.03768, 2017. [7] A. Menon, B. Rooyen, and N. Natarajan. Learning from binary labels with instance-dependent corruption. Machine Learning, 2018.

[Author Response · NeurIPS 2019]

We thank all the reviewers for the helpful reviews. We respond to each reviewer's specific questions here.

To Reviewer #1:

1. *The proposed $\mathcal{L}_{DMI}$ is not limited to DNN. Why does the paper focus on DNN practice?* Yes, $\mathcal{L}_{DMI}$ is not limited
to DNN and can be applied to general settings. Our theory is valid in the general settings. Our experimental setting
focuses on DNN practice since the noisy-labels problem in data-driven deep learning is especially important as large
high-quality data is crucial to data-driven deep learning but may be extremely hard to obtain (see intro), thus a series
of works [21, 11, 6, 36] also focus on designing noise-robust functions for DNN. Applying our method to non-DNN
settings can be a future direction but we believe our current results not only make a significant theoretical step in the
general noisy-label learning problem but also make a significant empirical step in the DNN practice.

2. *The sufficiency of the paper's literature review and experimental comparison are kind of weak.* We have compared
our method with some very recent baselines like **GCE** [36] in NIPS 2018, **LCCN** [33] in AAAI 2019 and also cited
several works that focus on network structure or learning paradigm (see the second and third paragraphs in related
work). We would appreciate if you could provide our missing works and will add them in our final version.

To Reviewer #2:

1. *Once the latent true variable $Y$ is not identifiable, how to find the optimal classifier for $Y$ ?* Yes, there exists $Y' \neq Y$
such that conditioning on $Y'$, $X$ is also independent of $\tilde{Y}$. $Y' = \tilde{Y}$ is one example. However, first, training with $Y$ or $Y'$
or $\tilde{Y}$ will obtain the same optimal classifier based on our main theorem. Second, based on the information-monotonicity
of $\mathcal{L}_{DMI}$, if $Y$ is more informative about $X$ than $Y'$ (e.g. when $Y' = \tilde{Y}$), the optimal classifier we obtained will be more
similar with $Y$ than $Y'$. When we assume the ground truth is the most informative among all variables that satisfy
the conditional independent assumption, the optimal classifier will be closest to the ground truth. We will add this
clarification in our final version.

2. *Why it doesn't work well on small noise?* We have the clean data comparisons since $r = 0$ is the clean case. **DMI**
outperforms all other counterparts except **GCE** [36] when $r = \{0.0, 0.1, 0.2\}$ in the Dogs vs. Cats dataset and when
$r = 0.0$ in CIFAR-10. **GCE** does a training optimization on **MAE** [6] with some hyperparameters while sacrifices the
robustness a little bit theoretically. Our next step may be to employ some training optimizations in our method.

3. *A Rate of ... papers.* Thanks for your information! We will cite them and make it clear in the final version.

To Reviewer #3:

1. *Why converting to two classes and other baselines are missing for Fashion-MNIST?* We use it as an explanation
experiment to compare distance-based (we use **CE** as an example here) and information-theoretic (our method) loss
functions (see the third paragraph in intro). For clean presentation, we convert to two highly imbalanced classes and
only compare with **CE**. We will make it clear in our final version. We attach the comparisons with other methods here
(see the following figures) and will include them in Appendix in our final version.

2. *Why are CIFAR-10 uniform, Dogs vs. Cats uniform and dog->cat missing?* Due to space limitation, we omit some
under-representative experiments although we have done them. For the uniform noise, first we want to clarify that noise
patterns are divided into two main cases, diagonally dominant and diagonally non-dominant and uniform noise is a
special case of diagonally dominant noise. Thus, we did not emphasize the uniform noise results in the submission. We
did not present the results for dog->cat since, unlike the highly imbalanced Fashion-MNIST (90% clothes, 10% bags),
Dogs vs. Cats is a balanced dataset (50% dogs, 50% cats) and thus the dog->cat results are very similar to cat->dog
results. We attach these results here (see the following figures) and will include them in Appendix in our final version.



[Meta-Review · NeurIPS 2019]

This paper proposed a novel information-theoretic loss function for learning with noisy labels. Besides regularizations borrowed from other areas, current directions in this area include loss correction, sample selection/reweighting, and label correction; different from existing loss-correction methods, the proposed loss is theoretically insensitive to label noise. This novelty is the reason for an acceptance recommendation. However, the authors didn't address well the reviewers' concerns in the rebuttal. I called the 4th reviewer to perform a quick review, and this reviewer is happy with the idea but again unhappy with the experiments done so far. Note that this acceptance is only conditional, so please carefully follow the reviews to revise the manuscript and prepare the final version.